# Detrimental role of IL-33/ST2 pathway sustaining a chronic eosinophil-dependent Th2 inflammatory response, tissue damage and parasite burden during *Toxocara canis* infection in mice

Thaís Leal-Silva[1,2], Flaviane Vieira-Santos[1], Fabrício Marcus Silva Oliveira[1], Luiza de Lima Silva Padrão[1], Lucas Kraemer[1], Pablo Hemanoel da Paixão Matias[1], Camila de Almeida Lopes[1], Ana Cristina Loiola Ruas[1], Isabella Carvalho de Azevedo[1], Denise Silva Nogueira[1], Milene Alvarenga Rachid[3], Marcelo Vidigal Caliari[3], Remo Castro Russo[4], Ricardo Toshio Fujiwara[1], Lilian Lacerda Bueno[1]*

1 Laboratory of Immunology and Genomics of Parasites, Department of Parasitology, Institute of Biological Sciences, Universidade Federal de Minas Gerais, Belo Horizonte, Brazil, 2 Post-graduation Program in Health Sciences: Infectious Diseases and Tropical Medicine, Faculdade de Medicina, Universidade Federal de Minas Gerais, Belo Horizonte, Brazil, 3 Laboratory of Protozooses, Department of General Pathology, Institute of Biological Sciences, Universidade Federal de Minas Gerais, Belo Horizonte, Brazil, 4 Laboratory of Pulmonary Immunology and Mechanics, Department of Physiology and Biophysics, Institute of Biological Sciences, Universidade Federal de Minas Gerais, Belo Horizonte, Brazil

* llbueno@icb.ufmg.br

## Abstract

Toxocariasis is a neglected disease that affects people around the world. Humans become infected by accidental ingestion of eggs containing *Toxocara canis* infective larvae, which upon reaching the intestine, hatch, penetrate the mucosa and migrate to various tissues such as liver, lungs and brain. Studies have indicated that Th2 response is the main immune defense mechanism against toxocariasis, however, there are still few studies related to this response, mainly the IL-33/ST2 pathway. Some studies have reported an increase in IL-33 during helminth infections, including *T. canis*. By binding to its ST2 receptor, IL-33 stimulating the Th2 polarized immune cell and cytokine responses. Thus, we aimed to investigate the role of the IL-33/ST2 pathway in the context of *T. canis* larval migration and the immunological and pathophysiological aspects of the infection in the liver, lungs and brain from Wild-Type (WT) BALB/c background and genetically deficient mice for the ST2 receptor (ST2$^{-/-}$). The most important findings revealed that the IL-33/ST2 pathway is involved in eosinophilia, hepatic and cerebral parasitic burden, and induces the formation of granulomas related to tissue damage and pulmonary dysfunction. However, ST2$^{-/-}$ mice, the immune response was skewed to Th1/Th17 type than Th2, that enhanced the control of parasite burden related to IgG2a levels, tissue macrophages infiltration and reduced lung dysfunction. Collectively, our results demonstrate that the Th2 immune response triggered by IL-33/ST2 pathway mediates susceptibility to *T. canis*, related to parasitic burden, eosinophilia and granuloma formation in which consequently contributes to tissue inflammation and injury.

**Data Availability Statement:** All relevant data are within the manuscript and its Supporting Information files.

**Funding:** This investigation received partial support from Fundação de Amparo a Pesquisa do Estado de Minas Gerais/FAPEMIG, Brazil (Grant# CBB APQ-00766-18), the Brazilian National Research Council (CNPq) (Grant# 421392/2018-5 and Grant# 302491/2017-1) and Pró-Reitoria de Pesquisa of Universidade Federal de Minas Gerais to cover research inputs. TLS is grateful for the PhD fellowship provided by the Brazilian National Research Council (CNPq), Post-graduation Program in Infectology and Tropical Medicine/ Universidade Federal de Minas Gerais. MCV, MAR, RCR, RTF and LLB are Research Fellows from the Brazilian National Research Council (CNPq). The funders had no role in study design, data collection and analysis, decision to publish, publication fees, or preparation of the manuscript.

**Competing interests:** The authors have declared that no competing interests exist.

## Author summary

Toxocariasis is a neglected disease caused by *Toxocara canis*, which has 19% worldwide seroprevalence, and is associated with socioeconomic, geographic and environmental factors. Humans become infected by accidental ingestion of *T. canis* eggs present in contaminated food, water or soil. After ingestion, the larvae hatch in the intestine and can reach various tissues such as liver, lung and brain. Helminth infections usually trigger a Th2 immune response in the host, by releasing cytokines such as IL-4, IL-5, IL-13 and IL-33. IL-33 is an alarmin that binds to the ST2 receptor, and some studies have observed an increase in this cytokine in toxocariasis, however there are no studies regarding the IL-33/ ST2 role in this infection. Thus, we evaluated the influence of this pathway by analyzing immunological and pathophysiological aspects in *T. canis*-infected mice. Our results demonstrated that the IL-33/ST2 pathway is related to parasite burden on the liver and brain and also increases the number of eosinophils in the blood and tissues. In addition, it involved with the pulmonary immune response and granulomas with impact in lung function. In conclusion, the IL-33/ST2 pathway governs the host susceptibility to *T. canis* in mice.

## Introduction

Toxocariasis is a zoonosis caused by nematodes of the genus *Toxocara*, whose main etiologic agent is *Toxocara canis* [1,2], a neglected disease with cosmopolitan distribution worldwide and seroprevalence rates estimated at 19%. Of these, the highest rates are associated with socioeconomic, geographic and environmental factors [3].

Humans are infected by accidental ingestion of *T. canis* eggs containing infective third stage larvae present in contaminated food, water, soil or utensils. After ingesting eggs, the larvae penetrate the intestinal mucosa and migrate to multiple organs [4]. Human toxocariasis can manifest in different ways based on the parasite tropism, and the severity of the disease will depend on the parasitic burden, the duration of larval migration, aging and immune-mediated responses of the immunocompromised individuals [2,5]. According to the larval migration site and clinical symptoms, toxocariasis is divided into four clinical forms: the Visceral larva migrans (VLM), Neurotoxocariasis (NT), Ocular toxocariasis (OT) and Covert or Common toxocariasis (CT) [2,4,5].

Experimental models have proven to be important to better understand the disease and mice have been widely used for this purpose [6–8], since the migration pathway, immune responses and lesions produced by *T. canis* larvae in humans and mice are similar [5,7–10]. It has been reported that during infection there is presence of persistent pulmonary inflammation, airway hyperreactivity and production of Th2 type cytokines, with the presence of eosinophilia and production of specific antibodies [4,11]. Some studies describe the presence of granulomas in liver lesions, with the participation of Th1 immune response and innate immunity cells, mainly eosinophils and macrophages [12,13]. In the brain, it is common to have hemorrhagic areas with recruitment of neutrophils, eosinophils and activation of microglia/ macrophages [6,14]. The excretory-secretory *T. canis* (TES) antigens normally stimulate the production of Th2 type cytokines such as IL-4, IL-5, IL-13 and IL-33, and there is an increase in eosinophils, IgE and IgG antibodies and Th1 cytokines reduction [4,6,8].

The ST2 receptor is a member of the IL-1 superfamily and is the IL-33 cytokine receptor [15]. Several tissues and innate immunity cells express IL-33, including macrophages,

dendritic cells, mast cells, epithelial, fibroblasts and glial cells [16]. The binding of IL-33 to its ST2 receptor activates transcription factors, via MyD88 dependent but TRIF independent pathway, and stimulates the production of Th2 cytokines such as IL-4, IL-5 and IL-13 in wild type mice, also restoring the Th2 asthma phenotype in IL-4 deficient mice [17].

The activation of the IL-33/ST2 pathway has been previously studied in nematode infection. In infection with *Strongyloides venezuelensis* it is responsible for inducing pulmonary eosinophilic inflammation, maintaining airway hyperresponsiveness, Th2 response in the lungs and consequently may play a role in expelling the worms from the lungs[18,19]. Resende *et al.* [6] observed high concentrations of IL-33 in BALB/c mice infected with *T. canis* from the 5th day after infection, however, the importance of activating the IL-33/ST2 pathway to control parasitic burden and pathophysiology in the context of *T. canis* infection has not yet been demonstrated.

Therefore, we believe that it is necessary to understand the immunological and pathophysiological mechanisms related to the IL-33/ST2 pathway during infection by *T. canis*. Thus, we further investigate if the IL-33/ST2 pathway may be relevant to the host's immune response during *T. canis* infection. Our results suggested that the IL-33/ST2 pathway in *T. canis* infection is important for the establishment of eosinophilia and its presence is related to hepatic and cerebral parasitic tropism, with increased liver and lung tissue inflammation, with loss of function. However, ST2$^{-/-}$ mice showed a skewed Th1/Th17 immune response with reduced parasite load, attenuating the tissue injury and chronical inflammation. Thus, the IL-33/ST2 pathway sustained the Th2 immune response contributing to eosinophil activity, tissue damage and parasite tropism during infection by *T. canis* in mice.

## Material and methods

### Ethics statement

All procedures performed during the experiments were conducted according to the Brazilian College of Animal Experimentation (COBEA) and approved by the local Animal Ethics Committee (CEUA) of the Federal University of Minas Gerais (UFMG), under protocol number 56/2018.

### Animals

For this study were used female BALB/c mice (*Mus musculus*) with approximately 8 weeks of age genetically deficient for the ST2 receptor (ST2$^{-/-}$), which were kindly provided by Dr. João Santana da Silva of University of São Paulo (USP) and wild-type mice (WT) BALB/c obtained from the animal facility of the Federal University of Minas Gerais.

During the experimental period, the mice were fed with filtered water and commercial chow (Nuvilab Cr-1, Nuvital Nutrients, Brazil) *ad libitum*. Mice were maintained at the Animal Facility of the Department of Parasitology of the Federal University of Minas Gerais under controlled conditions of temperature (24 ± 1˚ C) and lighting (12-hour light-dark cycle).

### Parasites

Adult *Toxocara canis* worms were obtained from the feces of naturally infected puppies which were kept at the Zoonosis Control Center (Belo Horizonte, Minas Gerais, Brazil). The puppies were treated with anti-helminthics (Drontal Puppy, São Paulo, Brazil) in the dosage of 1mL/kg. The adult parasites of *T. canis* were collected after elimination in fecal samples and maintained in water until being processed at the Laboratory of Immunology and Genomics of Parasites at the Federal University of Minas Gerais, Brazil.

The adult female worms were dissected and the uterus was removed. Purification of the eggs was performed as previously described [6]. Briefly, the eggs were isolated from the uterus

by mechanical maceration purified by filtration on 100μm nylon strainers. Then, they were incubated for embryonation in 50 mL culture flasks with 0.2 M $H_2SO_4$. The culture was kept in a BOD incubator at $26 \pm 1°C$, undergoing oxygenation three times per week by stirring. After 6 weeks of culture, fully embryonated eggs were used in experimental infections.

### Experimental infection

On the day of inoculation, the culture eggs were incubated with 5% sodium hypochlorite solution in a $CO_2$ incubator for 1 hour and 40 minutes to facilitate the hatching of the larvae, then the solution was washed in water to remove the sodium hypochlorite.

All mice, except for the control group (0dpi), were inoculated via the intra-gastric route by gavage with 1000 embryonated *T. canis* eggs, followed by 0.1 mL of $H_2O$ to rinse the remaining eggs from the syringe and needle [6].

### Experimental design

The WT and ST2$^{-/-}$ mice were randomized and euthanized on day 0 (control group), 3, 14, and 63 days post-infection (dpi) to evaluate acute and chronic phase of infection (Fig 1A). At all times of infection (dpi), including the control group, a single experiment was carried out with 13 animals, 7 for histopathological analysis and 6 for other analysis. All mice were euthanized with a lethal injection of xylazine/ketamine (8.5 mg/kg and 130 mg/kg).

### Parasitological analysis

The parasitic burden was assessed by the recovery of larvae from the liver, lungs and brain. Each tissue was collected and sliced with scissors and placed in a Baermann apparatus for 4 hours in the presence of PBS (0.4 M NaCl and 10 mM NaPO4) at 37°C. The recovered larvae were then fixed (10% formaldehyde in PBS) and counted in an optical microscope [6].

### Leukocyte analysis

Approximately 500μL of blood was collected by cardiac puncture and then transferred to tubes containing the EDTA anticoagulant for leukocyte analysis. Subsequently, the tubes were centrifuged to collect the plasma, which was stored at -80°C for later analysis. The global leukocyte count was performed using a Bio-2900 Vet automatic hematological counter. For differential white blood cell counting, blood smears were stained with Panotic (Laborclin, Brazil), and 100 white blood cells were differentiated under a light microscope.

### Bronchoalveolar lavage

Bronchoalveolar lavage (BAL) was performed by inserting a 1.7 mm catheter into the trachea of mice and washing the lungs twice with 1 ml of sterile PBS. BAL samples were centrifuged at $300 \times g$ for 10 min at 4°C and the pellet was used to determine the total and differential cellularity using optical microscopy as described [20]. Supernatants were used for hemoglobin and total protein quantification, according to the manufacturer's instructions.

Alveolar hemorrhage was assessed by quantifying the hemoglobin present in the BAL, using the Hemoglobin K023-1 kit (Bioclin Quibasa, Brazil). The concentration was determined by spectrophotometry by measuring the absorbance at 540 nm. Hemoglobin (Hb) content was expressed in g/dL of Hb per ml of BAL. The quantification of total protein was determined by the BCA Protein Assay kit (Thermo Scientific, USA) as described [21]. The results were expressed in μg of total protein per ml of BAL.

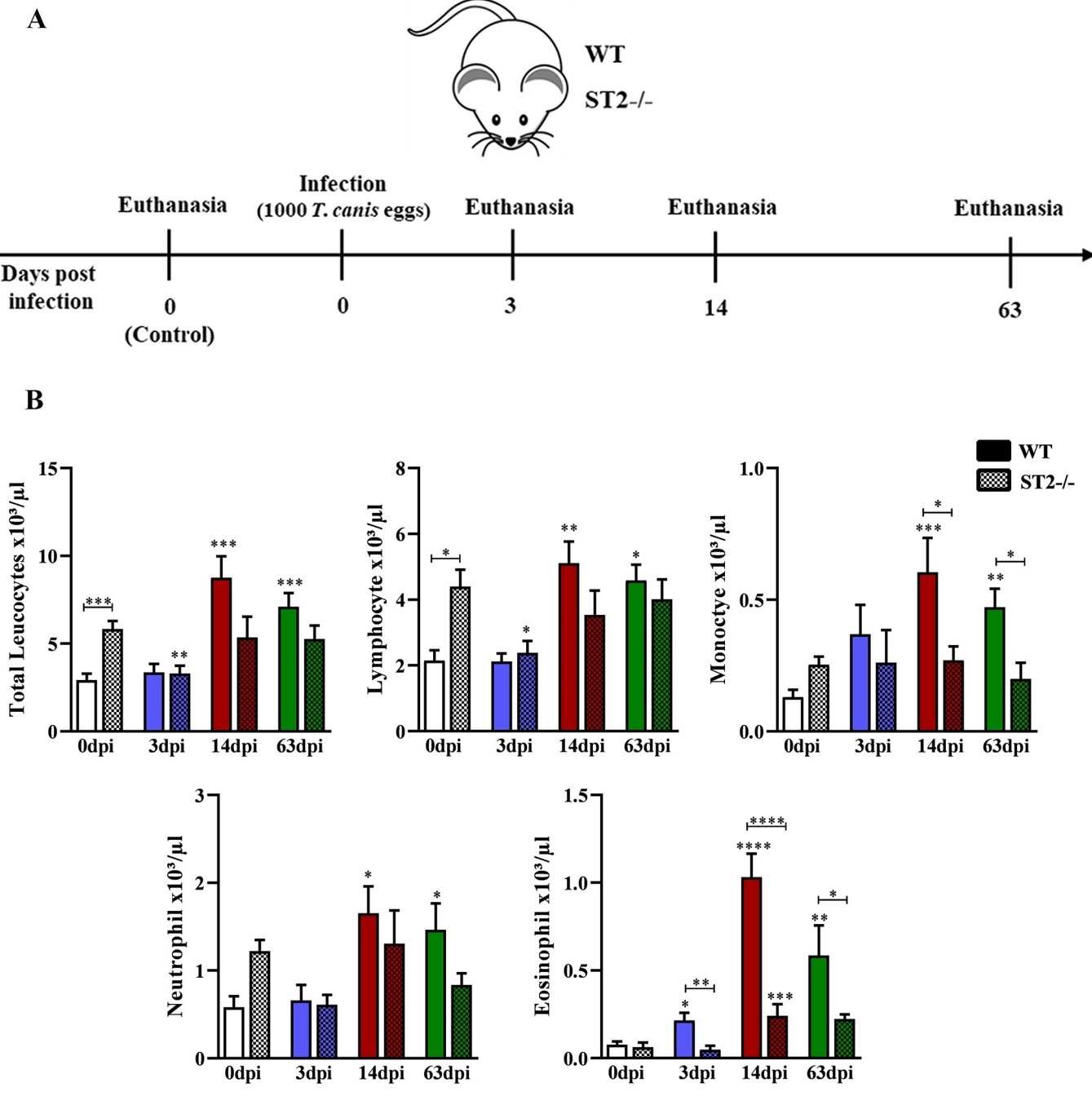

**Fig 1. Characterization of the experimental design and leukogram of WT and ST2$^{-/-}$ mice infected with *T. canis*.** (A) Experimental design of the infection with 1000 *T. canis* eggs at 0 (control group), 3, 14 and 63 days after infection; (B) Leukogram of different days after infection with *T. canis*. Statistical comparisons were made between each strain with its specific uninfected group (0dpi) represented by the asterisk without the bar and between strains at the same time of infection represented by the asterisk with the bar. Results represent mean ± S.E.M., *p<0.05, **p<0.01, ***p<0.001, ****p<0.0001. One-way ANOVA test and Kruskal-Wallis test followed by Dunn's test were used.

## Evaluation of enzymatic activity of alanine aminotransferase (ALT)

To assess liver enzyme activity, alanine aminotransferase (ALT) was quantified by means of a colorimetric enzymatic assay using the Transaminase ALT-K035-1 kit (Bioclin Quibasa, Brazil) determined by the manufacturer instructions.

## Pulmonary cytokine profile

To assess the concentration of pulmonary cytokines, the right lung of each animal was removed and homogenized (TissueLyser LT—Qiagen, German) in extraction solution (0.4 M NaCl, 0.05% Tween 20, 0.5% BSA, 0.1 mM phenylmethylsulphonyl fluoride, 0.1 mM benzethonium chloride, 10 mM EDTA and 20 IU of aprotinin A) at the rate of 1 mL per 100 mg of lung tissue. The homogenates were centrifuged at $800 \times g$ for 10 min at 4˚C, and the supernatant was collected and stored at −80˚C for cytokine quantification. Levels of IL-1β, TNF-α, IFN-γ, IL-12/IL-23p40, IL-6, IL-4, IL-33, IL-13, IL-5, IL-10, TGF-β and IL-17A were assayed by sandwich ELISA kit (R&D Systems, USA) according to the manufacturer's instructions. The absorbance of the samples was determined using a Versa Max ELISA microplate reader (Molecular Devices, USA) at a wavelength of 492 nm.

## Eosinophil peroxidase (EPO) and macrophage n-acetylglucosaminidase (NAG) assays

The indirect activity of macrophages and eosinophils was assessed by the concentration of N-acetylglucosaminidase (NAG) and eosinophils peroxidase (EPO), respectively, in pulmonary and brain homogenates performed according to the method previously described [21,22]. After tissue homogenization (TissueLyser LT-Qiagen, Hilden, Germany), the homogenate was centrifuged at $800 \times g$ for 10 min at 4˚C, and the resulting pellet was used to determinate NAG and EPO activity. Absorbance was expressed by a VersaMax ELISA Microplate Reader (Molecular Devices, USA) according to each assay and the results were expressed as optical densities (O.D.).

## Extraction of crude antigen from L3 larva

For the production of crude larvae L3 antigen, the culture eggs were centrifuged at $800 \times g$ for 10min at room temperature (RT) and incubated with 5% sodium hypochlorite solution in a $CO_2$ incubator for 1 hour and 40 min. After incubation, the solution with embryonated eggs was concentrated and washed three times in 20 mL PBS by centrifugation at $800 \times g$ for 10 min at RT.

The supernatant was discarded and Hank's saline solution (HBSS), pH 2.0, was added to the pellet containing the embryonated eggs and incubated for 30 min in 5% CO2 incubator at 37˚C. After that period, HBSS at pH 7.0 was added. Then, the embryonated eggs were centrifuged at $800 \times g$ for 10 minutes at RT and the solution was resuspended in RPMI-1640 medium (SIGMA, USA), supplemented with 4% penicillin/streptomycin (Invitrogen, USA) and placed in 24-well plates at 37˚C and 5% CO2 for a period of 72 hours for the larvae to hatch. After incubation, L3 larvae were collected and transferred to a 50 ml graduated tube, where they were centrifuged at $800 \times g$ for 10 min at RT. The supernatant was discarded and the pellet was resuspended in 5mL of PBS and was taken to a sonicator for 10 cycles of 1 min with an interval of 30 seconds each. After sonication, the contents were centrifuged at $800 \times g$ for 15 min at 4˚C, the supernatant was collected and stored at −80˚C until use. The amount of protein was measured using a commercial BCA kit (Thermofisher Scientific, USA), performed according to the manufacturer's instructions.

## Antibody detection

To measure the total IgG antibodies, ELISA assays were performed using the plasma of *T. canis* infected and non-infected animals. The assay was performed as previously described [23]. Briefly, the ELISA plates (Greiner-Bio-One, USA) were sensitized with 100μL of crude L3

larvae antigen diluted in carbonate bicarbonate buffer at a concentration of 1μg per well and left overnight at 4˚C. The next day, the plates were washed with the washing solution (PBS-0.05% Tween 20) and 250μL of the blocking solution (PBS + 3% BSA) was added, being incubated for 1 hour at 37˚C. After blocking, all solution from the wells was removed by aspiration, and 100μL of plasma were added to the wells, diluted 1:1000 in PBS with 3% BSA and incubated at 4˚C overnight. The next day, the plates were washed with the washing solution and 100μL of the anti-mouse IgG antibody conjugated to peroxidase diluted 1:2000 in PBS-BSA 3% was added. After incubation at 37˚C for 1 hour, the plates were washed again, and 100μL of the developer solution (0.1 M citric acid, 0.2 M $Na_2PO_4$, 0.05% OPD and 0.1% $H_2O$) was added. The plates were incubated at 37˚C in the dark for 20 minutes and the reaction was stopped by adding 50μL of 0.2 M $H_2SO_4$. Absorbance was measured in an ELISA reader with a absorbance of 492 nm. All tests were performed in duplicates. The ELISA assay for IgE, IgG1, IgG2a, IgG2b and IgG3, was the same as described above, except for the dilutions of plasma that were 1:10, 1:200, 1:100, 1:100 and 1:200 respectively, and the secondary antibodies anti-IgE mouse, anti- IgG1 mouse, anti- IgG2a mouse, anti- IgG2b mouse and anti-IgG3 mouse, which were diluted in PBS-BSA 3% in 1:1000, except anti-IgG2a mouse which was diluted 1:500.

## Histopathological and morphometric analysis

After removal of the organs, the left lung, the right lobe of the liver and the left half of the brain were fixed in a buffered 10% formaldehyde solution (Synth, Brazil) for seven days. Subsequently, the samples were gradually dehydrated in ethanol, diaphanized in xylol and included in paraffin blocks, obtaining 4μm thick cuts with which histopathological slides were made and stained with hematoxylin and eosin (HE) for histopathological, semiquantitative morphometric analysis. All histopathological analysis were performed blindly.

For the liver lesions score, 10 random images were captured per animal with a 20X magnification. The score was based on 4 degrees for hepatic parenchyma injury: grade 0, absence of inflammatory cells; grade 1, some regions of the liver parenchyma had small inflammatory foci with reduced number of inflammatory cells, small areas of necrosis; grade 2, the hepatic parenchyma had inflammatory foci with a moderate number of cells, perivascular inflammatory infiltrate, as well as around the ducts and small areas of necrosis dispersed by the parenchyma; grade 3, hepatic parenchyma frequently presented larger inflammatory foci, diffuse inflammatory infiltrate, exuberant perivascular inflammation and around the ducts, areas of necrosis dispersed by the parenchyma. Counting of hepatic and pulmonary granulomas was also performed.

For the airway inflammation score were captured 10 random images per animal with 20X magnification, as described by [23] and analyzed perivascular inflammation, peribronchial inflammation, parenchyma injury, and hemorrhage.

Morphometric analysis of liver injury severity was performed by calculating all areas of inflammation and necrosis, while in the brain the hemorrhagic areas were evaluated, as described by [6,24].

The degree of thickening of the interalveolar septa was assessed by capturing thirty random images with a 40X objective, comprising an area of $1.6 \times 10^6$ μm$^2$. Using the KS300 software, all pixels of the lung tissue in the real image were selected for the creation of a binary image, digital processing and calculation of the area in μm$^2$ of the interalveolar septum. The analysis of the pulmonary inflammatory infiltrate was performed on the images previously selected to assess the thickening of the interalveolar septa. All cells contained in each image were quantified using the Carl Zeiss Image Analyzer KS300 software program [25].

### Assessment of respiratory mechanics

The evaluation of pulmonary function was carried out in mice WT and ST2$^{-/-}$ infected with 0dpi (control group) and 3dpi (peak of larvae in the lung) as described by [20,23,26]. Briefly, the animals were anesthetized by a subcutaneous injection (8.5 mg/kg xylazine and 130 mg/kg ketamine) to maintain spontaneous breathing and then were tracheostomized and placed on a body plethysmograph, connected to a computer-controlled ventilator (Forced Pulmonary Maneuver System, Buxco Research Systems, Wilmington, Carolina do North, USA). First, an average breathing frequency of 160 breaths/min was imposed to the animal by pressure-controlled ventilation. Under mechanical respiration the Dynamic Compliance (Cdyn), the Tidal Volume (TV) and Lung Resistance (Rlung) were determined by Resistance and Compliance (RC) test. To measure the Chord Compliance (Cchord) the quasi-static pressure-volume maneuver was performed, which inflates the lungs to a standard pressure of +30 cmH2O and then slowly exhales until a negative pressure of -30 cmH2O is reached. Chord compliance was determined at 10 cmH2O pressure. Fast-flow volume maneuver was performed, and the lungs were inflated to +30 cmH2O, and afterwards immediately connected to a highly negative pressure in order to enforce expiration until -30 cmH2O. The Forced Vital Capacity (FVC), Forced Expiratory Volume at 50 milliseconds (FEV50), and the Flow-Volume Curves were recorded during this maneuver. Suboptimal maneuvers were rejected, and for each test in every single mouse, at least three acceptable maneuvers were conducted to obtain a reliable mean for all numeric parameters.

### Statistical analysis

The Prism 8.0 software (GraphPad Inc, USA) was used for the statistical analysis. Grubb's test was used to detect possible outlier values. The Shapiro-Wilk test was performed to verify the distribution of the data. For the comparison between the uninfected groups (0dpi) and each days of infection, one-way ANOVA test followed by Tukey's multiple comparisons test or the Kruskal-Wallis non-parametric test followed by Dunn's post-test were used. All tests were considered significant when the p value was equal to or less than 0.05.

## Results

### Absence of the IL-33/ST2 pathway reduces eosinophilia and increases antibodies levels

There are few data in the literature that evaluate the role of IL-33/ST2 pathway in helminth infections, and none of them assess its role during infection by *T. canis*, so we sought to determine whether the IL-33/ST2 pathway influences immunological and pathophysiological parameters during infection by *T. canis*.

Initially, to assess the influence of the IL-33/ST2 pathway on the systemic aspects throughout the infection by *T. canis*, the leukocyte profile and the levels of antibodies in the peripheral blood were evaluated (Fig 1B). We observed that WT mice presented leukocytosis in the 14dpi and 63dpi with increased number of lymphocytes, monocytes, neutrophils and eosinophils, while ST2-/- mice showed leukocytosis only in the 3dpi with increased lymphocytes and eosinophilia in the 14dpi. Interestingly, comparing the two strains, uninfected ST2 mice have a higher number of peripheral leukocytes, especially lymphocytes when compared to WT. Furthermore, there was a reduction in the number of eosinophils and monocytes in ST2-/- mice with 14dpi and 63dpi. When analyzing the levels of antibodies, we observed that the total IgG concentrations during infection increased with 14dpi and 63dpi in both strains, with an increase in IgG1 and IgG3 in both strains, and an increase in IgG2a and IgG2b in ST2-/- mice

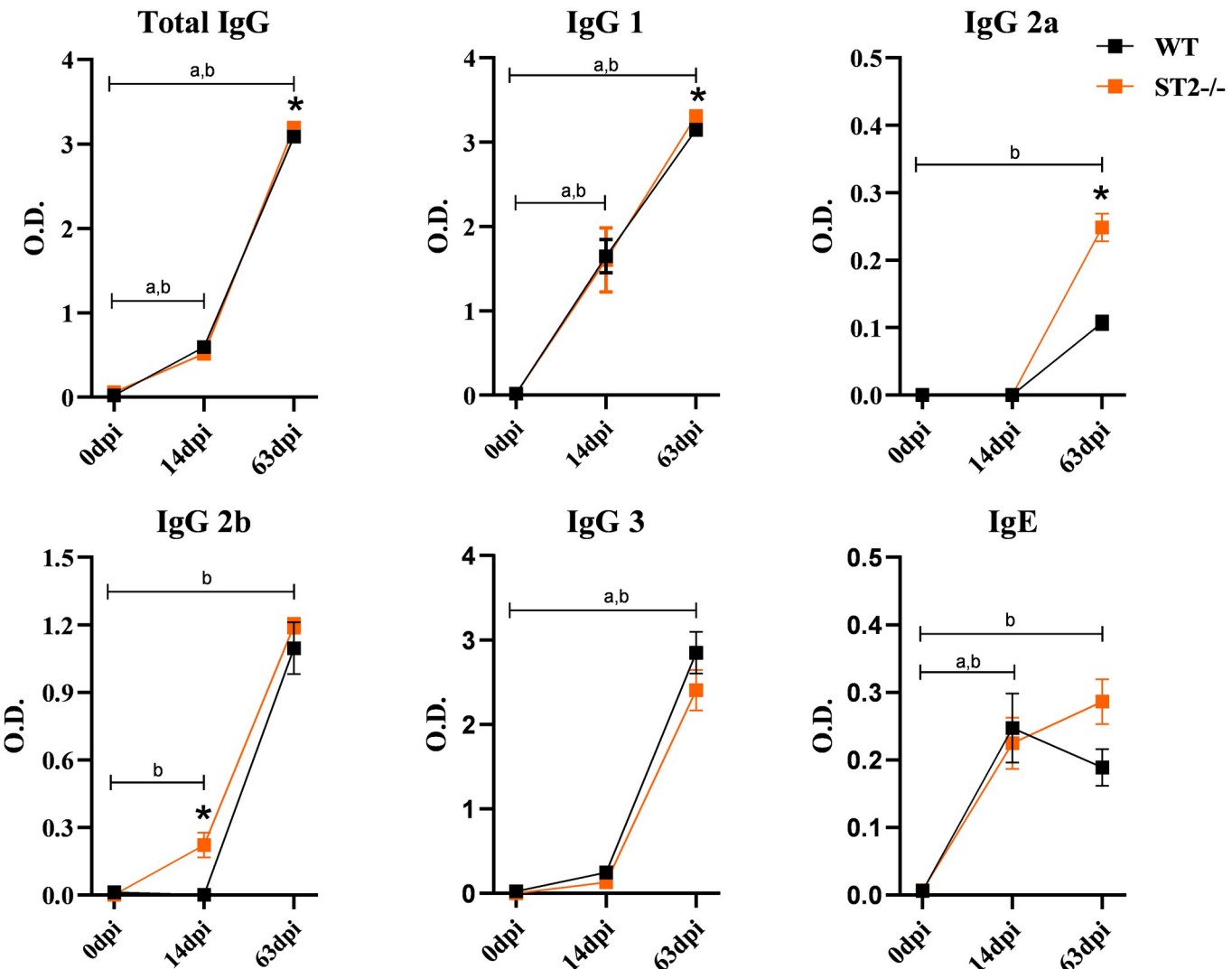

**Fig 2. Characterization of the humoral response against antigens from crude extract of L3 larvae in WT and ST2⁻/⁻ mice.** Evaluation of total IgG, IgG1, IgG2a, IgG2b, IgG3 and IgE antibodies against antigens from *T. canis* larvae in mice. Results represent mean ± S.E.M., *p<0.05; [a]Significant difference in the 0dpi WT group; [b] Significant difference in the 0dpi ST2⁻/⁻ group. One-way ANOVA test and Kruskal-Wallis test followed by Dunn's test were used.

with 63dpi compared to uninfected mice (Fig 2). When comparing the strains, we observed an increase in the concentration of IgG2b with 14dpi and IgG1 and IgG2a with 63dpi in ST2-/- mice compared to WT. As expected, IgE levels increased in both strains during infection. The results demonstrate that, at the systemic level, the IL-33/ST2 pathway is important for the establishment of eosinophilia and for the increase of monocytes in *T. canis* infection, in addition to contributing to the production of IgG1, IgG2a and IgG2b antibodies.

## Lack of ST2 receptor contributes to the reduction of parasitic load in the liver and tissue damage

We assessed the role of the IL-33/ST2 pathway in liver changes caused by *T. canis* infection. The ST2⁻/⁻ mice showed an intense reduction of parasitic burden and ALT enzyme with 3dpi (Fig 3A and 3B). In the analysis of the score, we observed a decrease in the granuloma count, which probably occurred due to the reduction of the parasitic load in the ST2⁻/⁻ mice, but did

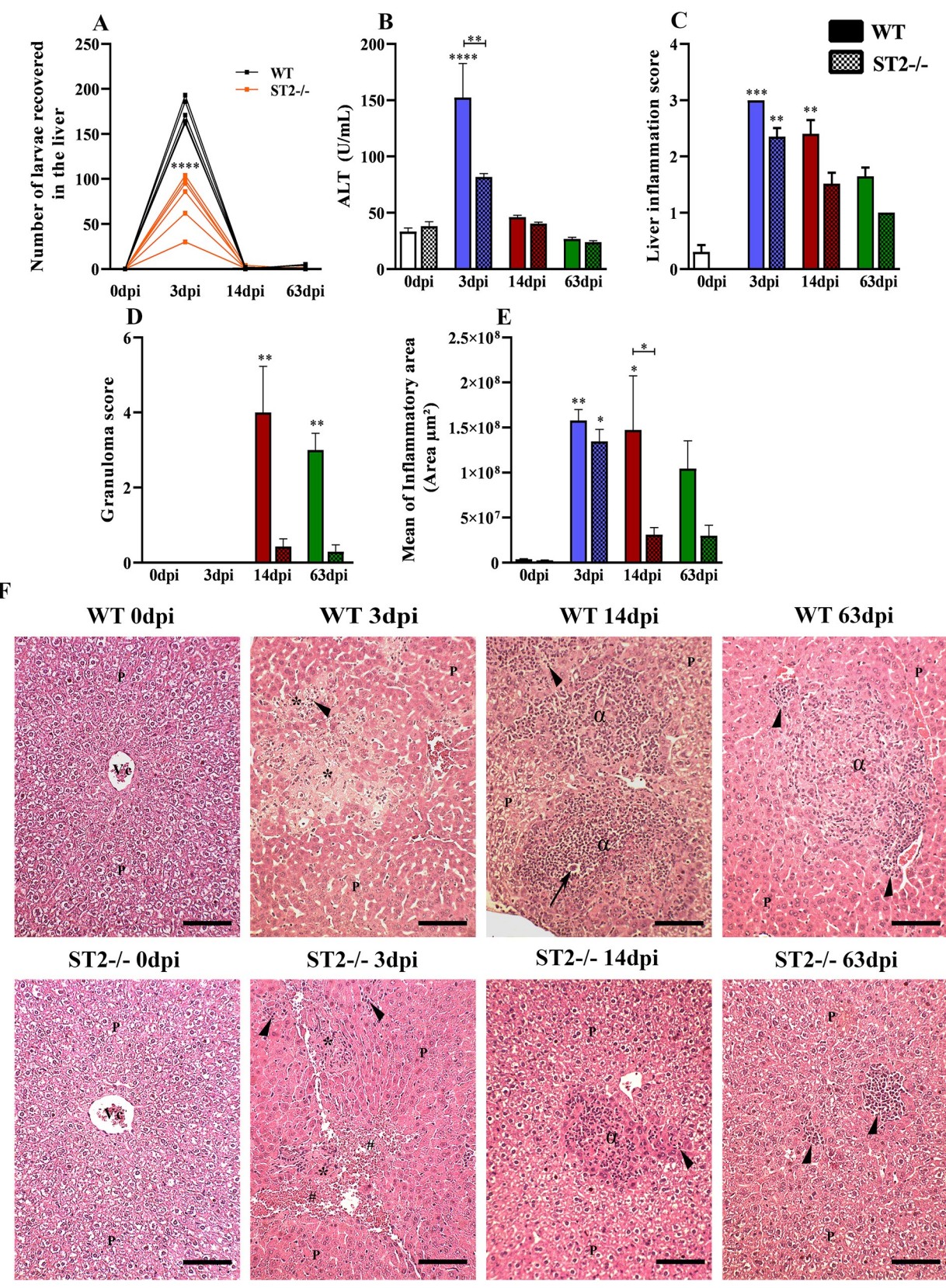

**Fig 3. Liver changes in WT and ST2$^{-/-}$ mice infected with *T. canis*.** (A) Number of larvae recovered from the liver; (B) Analysis of the enzymatic activity of plasmatic alanine aminotransferase (ALT); (C) Number of granulomas; (D) Analysis of liver inflammation by score; (E) Morphometric analysis of liver inflammatory areas; (F) Representative hematoxylin and eosin staining of liver sections, hepatic parenchyma (P), vein lobular center (Vc), inflammatory infiltration foci (arrowheads), necrosis area (*), hemorrhage area (#), granuloma (α), toxocara larvae (arrow). Bar = 200μm. Statistical comparisons were made between each strain with its specific uninfected group (0dpi) represented by the asterisk without the bar and between strains at the same time of infection represented by the asterisk with the bar. Results represent mean ± S.E.M., *$p<0.05$, **$p<0.01$, ***$p<0.001$, ****$p<0.0001$. One-way ANOVA test and Kruskal-Wallis test followed by Dunn's test were used.

not alter the liver inflammation (Fig 3C and 3D). In the morphometric analysis, a reduction in the hepatic inflammatory area was also observed in ST2$^{-/-}$ mice, confirming histopathological findings (Fig 3E). The histopathological analysis of the liver showed that in uninfected mice, the hepatocytes maintained a morphological aspect consistent with normality (Fig 3F). At 3dpi in both strains there were the presence of dispersed larvae in the parenchyma, larger areas of necrosis, hemorrhage and congestion of blood vessels and capillaries. It was also observed inflammatory foci characterized as mixed composed of eosinophils, neutrophils, macrophages and lymphocytes, however in ST2$^{-/-}$ mice the presence of eosinophils was scarce. At 14dpi, both strains continued to show inflammatory foci in the parenchyma, but in ST2$^{-/-}$ mice they were few and smaller with scarce eosinophils and lymphocytes. In both strains of mice, the presence of granulomas was also observed, which were in the exudative phase, mostly composed by eosinophils, followed by macrophages, necrotic-exudatives and necrosis zones, and larvae could be identified next to the granulomas, however in the ST2$^{-/-}$ mice the presence of granuloma was found only in 3 mice. At 63dpi the WT and ST2$^{-/-}$ mice continued to present inflammatory foci in the hepatic and perivascular parenchyma, with few eosinophils in the latter. The animals also presented granulomas in the exudative phase and granulomas in the productive phase composed of macrophages, giant cells, epithelioid cells, fibroblasts and collagen. However, in ST2$^{-/-}$ mice, only 2 animals presented granulomas. In both strains, vascular congestion was frequently observed throughout the liver parenchyma. Collectively, these data suggest that IL-33/ST2 pathway can influence the initial phase of migration of the larvae to the liver, increasing the parasitic load and, consequently, damage to the liver parenchyma.

## The pulmonary cytokine profile shows skewed Th1/Th17 immune response in ST2$^{-/-}$ mice during *T. canis* infection

After reaching the lungs, the larvae trigger an immunological response in the lungs, therefore we analyzed the influence of the IL-33/ST2 pathway on the pulmonary cytokine profile (Fig 4). We observed that infected WT mice exhibit a mixed response of Th1 and Th2 already in the initial stage of infection (3dpi), represented by an increase in IFN- γ, IL-1β, IL-33, IL-13 and IL-5. While in infected ST2$^{-/-}$ mice, only an increase in IL-33 was observed with 3dpi, which was expected due to the absence of the ST2 receptor. At 14 dpi, the immune response in both strains became mixed, with cytokines from the Th1, Th2 and Treg responses in WT mice and with Th1, Th2, Th17 and Treg cytokines in ST2$^{-/-}$ mice. Comparing the two strains, we observed an increase in IL-33 and a reduction in IL-5 with 3dpi, an increase in IL-17 and IL-13 with 14dpi, and with 63dpi increase in IL-33, IL-17 and IL-1β in ST2$^{-/-}$ mice when compared to WT. Thus, the results demonstrate that in *T. canis* infection induced a Th2 type of immune response via IL-33/ST2 pathway, however, the absence of ST2 receptor favoring a Th1/Th17 polarized immune response.

## ST2$^{-/-}$ mice show reduced lung inflammation and dysfunction during *T. canis* infection

Following the larval migration, the next step was to analyze the influence of the IL-33/ST2 pathway in the lungs during infection, through parasitic burden, tissue and airway

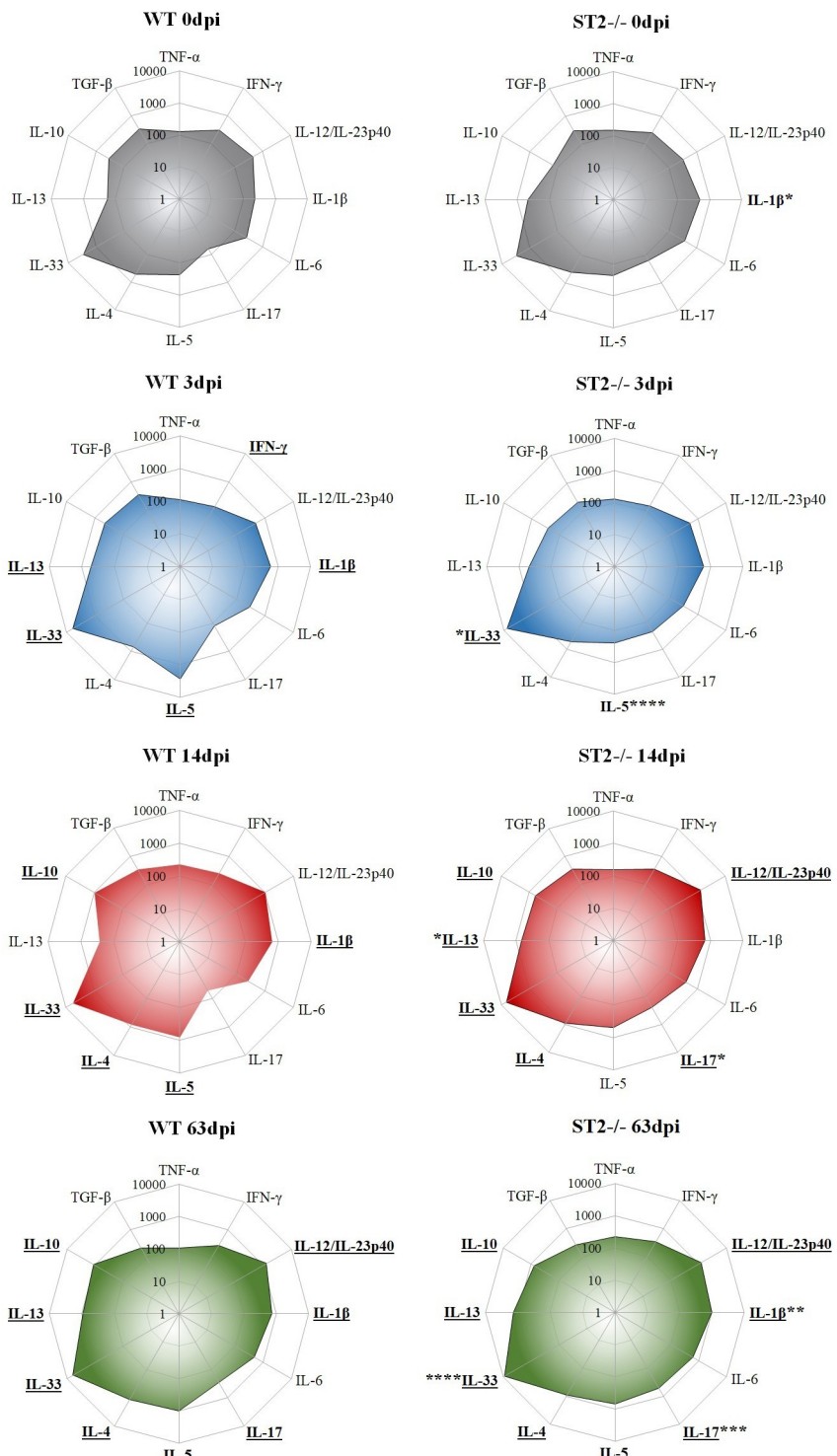

**Fig 4. Profile of cytokines present in lung tissue during infection with *T. canis*.** Radar graphs express the concentration of pulmonary cytokines throughout the infection. The results represent the mean ± S.E.M., $^*p<0.05$, $^{**}p<0.01$, $^{***}p<0.001$, $^{****}p<0.0001$ when compared to the corresponding WT groups. Cytokines were underlined when they had statistical difference with their respective control group (0dpi). One-way ANOVA test and the Kruskal-Wallis test followed by the Dunn test were used.

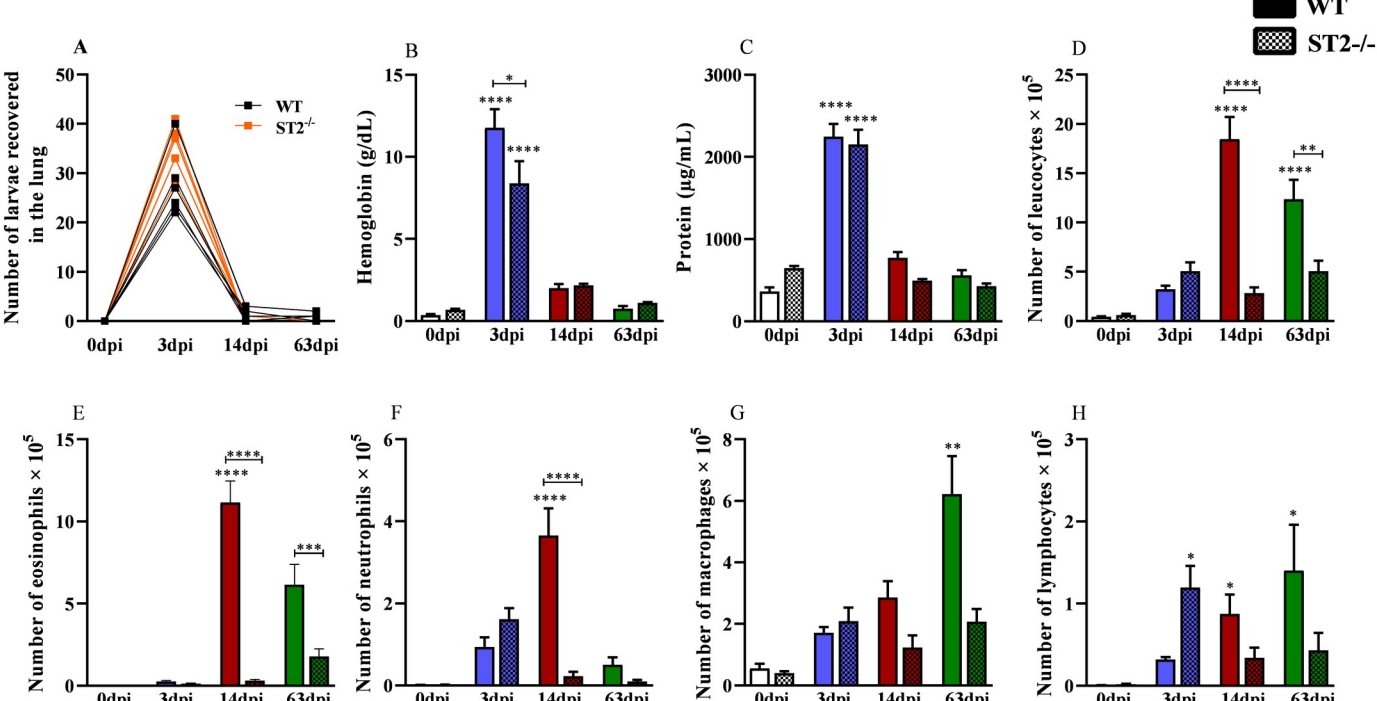

**Fig 5. Quantification of recovered larvae in the lungs and inflammation in the airways.** (A) Number of larvae recovered from the lung; (B) Hemoglobin quantification in the bronchoalveolar lavage (BAL); (C) Total protein levels in BAL; (D) Quantification of total leukocytes infiltration in BAL; (E) Number of eosinophils in BAL; (F) Number of neutrophils in BAL; (G) Number of macrophages in BAL; (H) Number of lymphocytes in BAL. Statistical comparisons were made between each strain with its specific uninfected group (0dpi) represented by the asterisk without the bar and between strains at the same time of infection represented by the asterisk with the bar. Results represent mean ± S.E.M., $^*p<0.05$, $^{**}p<0.01$, $^{***}p<0.001$, $^{****}p<0.0001$. One-way ANOVA test and Kruskal-Wallis test followed by Dunn's test were used.

inflammation, and pulmonary physiology. In both strains, the largest number of larvae was recovered in the 3dpi, with no significant difference between them at any time of the infection (Fig 5A). In the BAL were observed an increase in hemoglobin and protein with 3dpi and 14dpi in both strains, which may also be related to increased larval migration, however in the ST2$^{-/-}$ mice hemoglobin and protein levels were lower compared to WT (Fig 5B and 5C). When analyzing the leukocytes present in the BAL (Fig 5D, 5E, 5F, 5G and 5H), the results showed decrease in the total number of leukocytes with 14dpi and 63dpi in ST2$^{-/-}$ mice (Fig 5D), mainly with a reduction in eosinophils, neutrophils and macrophages in relation to the WT mice (Fig 5E, 5F and 5G).

When analyzing lung inflammation, we found an increase in NAG and EPO in both strains during infection, however there was an increase in NAG and a decrease in EPO with 3dpi and 14dpi in ST2$^{-/-}$ mice when compared to WT (Fig 6A and 6B). Histopathological scores (Fig 6C, 6D, 6E, 6F, 6G and 6H) and morphometric analysis (Fig 6I and 6J) in both strains showed hemorrhagic areas, inflammation and consequently an increase in the number of cells and thickening of the pulmonary septum, however, we observed a reduction in perivascular inflammation with 14dpi (Fig 6D), number of granulomas with 14dpi and 63dpi (Fig 6H) and the number of cells with 14dpi (Fig 6J) in ST2$^{-/-}$ mice, corroborating with the BAL data.

In the histopathological analysis of the pulmonary parenchyma (Fig 6K), we describe the lesions caused by the larval migration of the parasite, in terms of tropism, inflammatory infiltrate, presence or absence of larvae, granulomas, vascular and exudative phenomena. The uninfected mice exhibited a morphological pulmonary appearance consistent with normality.

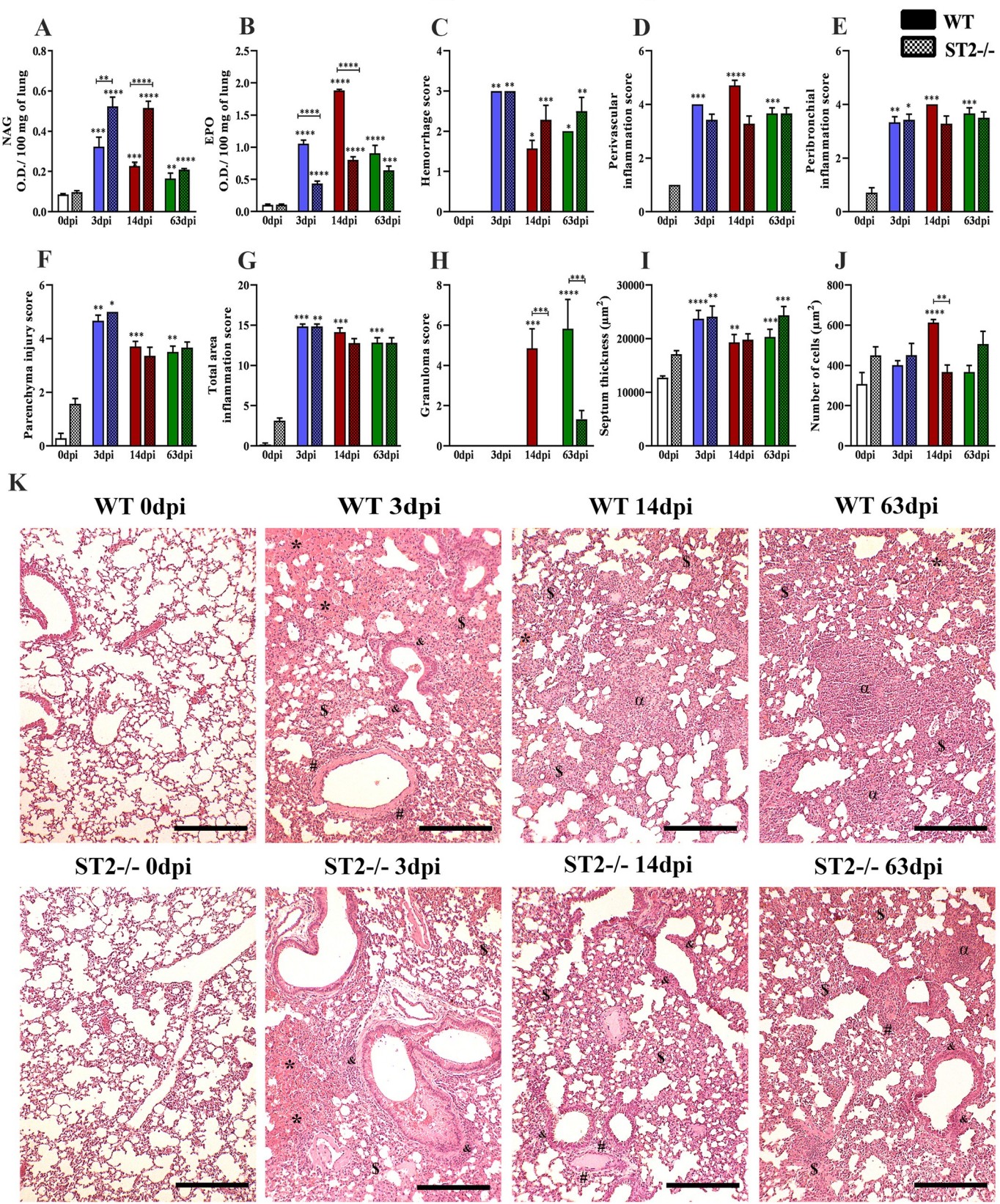

**Fig 6. Characterization of inflammation in the lung parenchyma in WT and ST2$^{-/-}$ mice infected with *T. canis*.** (A) N-acetylglucosaminidase (NAG) activity in lung tissue; (B) Eosinophils peroxidase (EPO) activity in lung tissue; (C) Hemorrhage score; (D) Perivascular inflammation score; (E) Peribronchial inflammation score; (F) Parenchyma injury score; (G) Total area inflammation score; (H) Granuloma score; (I) Septum thickness; (J) Number of cells; (K) Representative hematoxylin and eosin staining of lung sections, hemorrhage area (*), parenchyma inflammation ($), airways inflammation (&), vascular inflammation (#), granuloma (α), Bar = 400μm. Statistical comparisons were made between each strain with its specific uninfected group (0dpi) represented by the asterisk without the bar and between strains at the same time of infection represented by the asterisk with the bar. Results represent mean ± S.E.M., *$p < 0.05$, **$p < 0.01$, ***$p < 0.001$, ****$p < 0.0001$. One-way ANOVA test and Kruskal-Wallis test followed by Dunn's test were used.

With 3dpi in both strains, thickening of the interalveolar septa with mixed inflammatory infiltrate was observed, characterized by eosinophils, neutrophils, macrophages and lymphocytes. Exudative phenomena, such as perivascular edema, extensive hemorrhagic areas, dispersed larvae in the lung parenchyma, were also evidenced and hypertrophy of the cells of the bronchi and bronchioles was frequently observed. However, in the WT mice, granulomas were observed in the exudative phase, composed mostly of eosinophils and macrophages followed by lymphocytes. At 14dpi, the histopathological analysis was similar with 3dpi, however, in ST2$^{-/-}$ mice was observed diffuse inflammatory infiltrate with the scarce presence of neutrophils and eosinophils. In both strains, some macrophages had a brownish pigment in their cytosol suggestive of hemosiderin. In WT mice, the presence of granuloma in the exudative phase was frequently observed, not being found in ST2$^{-/-}$ mice. At 63dpi, both strains continued to show thickening of the interalveolar septa with diffuse inflammatory infiltrate, characterized as mixed in addition to the formation of Bronchus-Associated Lymphoid Tissue (BALT). It was also possible to show the presence of exudative and vascular phenomena in all mice, such as small hemorrhagic foci and capillary congestion. In all WT mice, there were granulomas in the productive phase in the lung parenchyma, while in ST2$^{-/-}$ mice the number of granulomas was reduced. These results indicate that the IL-33/ST2 pathway may be responsible for the increase of inflammation in the airways, mainly of eosinophils, in addition to favoring the formation of granulomas during *T. canis* infection.

## Disruption of IL-33/ST2 pathway attenuated lung dysfunction during infection by *T. canis*

To assess pulmonary dysfunction caused by *T. canis*, analysis of pulmonary mechanics was performed by spirometry at the peak of larval migration in the lungs in the 3dpi. (Fig 7). It was observed that *T. canis* infection contributes to the loss of the respiratory flow and volumes in consequence to tissue edema, as indicated by the reduction in the values of FVC and Flow, also with reduced pulmonary elasticity as indicated by increased lung resistance (Rlung) and reduced compliance (Cchord and Cdyn), besides causing changes in the airway flow with decreasing FEV50 in both strains. However, the ST2$^{-/-}$ mice had an improvement in relation to TV, Flow and FEV50, suggesting that the IL-33/ST2 is detrimental for lung injury and dysfunction during *T. canis* infection.

## Lack of the IL-33/ST2 pathway is related to the reduction of parasitic load and eosinophilic activity in the brain

After the larvae passed through the lungs, we evaluated their migration to the brain. The results of larvae recovery in brain tissue showed that the ST2$^{-/-}$ mice had a lower parasitic burden with 63dpi compared to the WT mice (Fig 8A). In both groups, lower levels of NAG were observed with 3dpi (Fig 8B). In addition, corroborating the lung data, ST2$^{-/-}$ mice also had a lower concentration of EPO in the 63dpi (Fig 8C). Analysis of the morphometry of the cerebral hemorrhagic area showed an increase in hemorrhage in both strains with 3dpi and in the WT mice with 63dpi (Fig 8D). Histopathological analysis of the brain of uninfected mice showed a

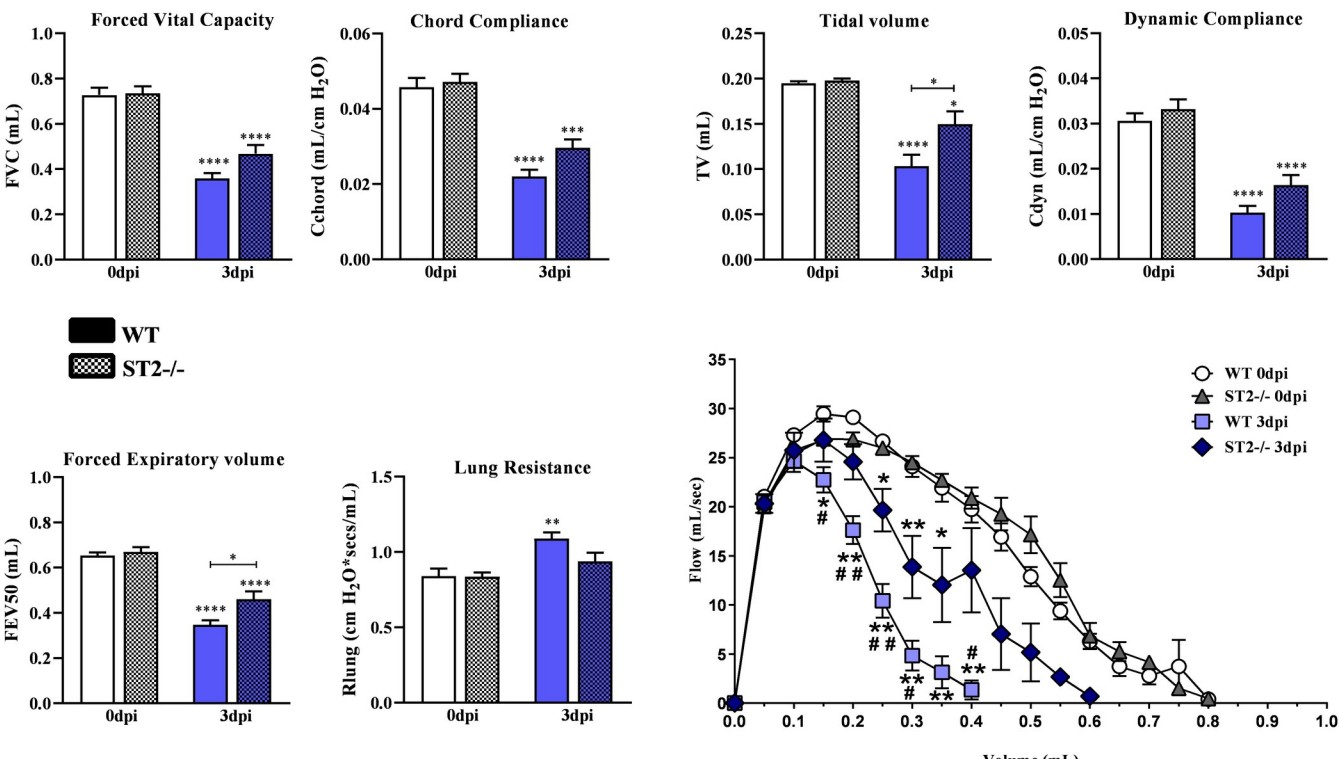

**Fig 7. Respiratory function analysis of WT and ST2$^{-/-}$ mice infected with *T. canis* with 3dpi.** Variables from lung mechanics were quantified: Forced vital capacity, Chord compliance, Tidal Volume, Dynamic compliance, Forced Expiratory Volume, Lung resistance and Flow–volume curve. Results represent mean ± S.E.M., $^*p<0.05$, $^{**}p<0.01$, $^{***}p<0.001$, $^{****}p<0.0001$, when compared with its specific uninfected group (0dpi) represented by the asterisk without the bar and between strains at the same time of infection represented by the asterisk with the bar; $\#p<0.05$ when compared to infected group, $\#\#p<0.001$ when compared to infected group. One-way ANOVA test was used.

morphological aspect consistent with normality, without histopathological changes (Fig 8E). With 3dpi, both strains demonstrated multifocal areas of hemorrhage and vacuolization of the neuropile in the cerebral cortex. Hemorrhagic foci were visualized in the cerebellum, located mainly in the white matter of the cerebellar folia and also in the hippocampus. Focal inflammatory infiltrates with lymphocytes, macrophages and occasional eosinophils were seen in the cerebral cortex. At 14 dpi, we observed occasional hemorrhagic foci in the cerebrum in both strains. And with 63dpi we observed in both groups that the white matter of the cerebellar folia had focal areas filled with "gitter cells" adjacent to the vacuolization areas and focal areas of gliosis and reactive blood vessels were detected in the cerebellum and cerebrum. Larvae and hemorrhagic foci were present in the cerebrum and hippocampus. Occasional perivascular accumulation of hemosiderophages in the cerebrum was observed. Together, the results showed that the IL-33/ST2 pathway contributes to the increase in eosinophil influx into the brain tissue at 63dpi, which may be related to increased cerebral parasitic load.

## Discussion

Studies show that several cells of the immune system express the ST2 receptor, such as Th2 cells, ILC2, cytotoxic T cells, B cells, macrophages, eosinophils, neutrophils and dendritic cells [27,28]. The total absence of the ST2 receptor, as shown in the animals used in our study, can deregulate the production of cytokines by macrophages, reduce the recruitment of immune cells to the site of infection and thus cause changes of Th2 response that may be important for

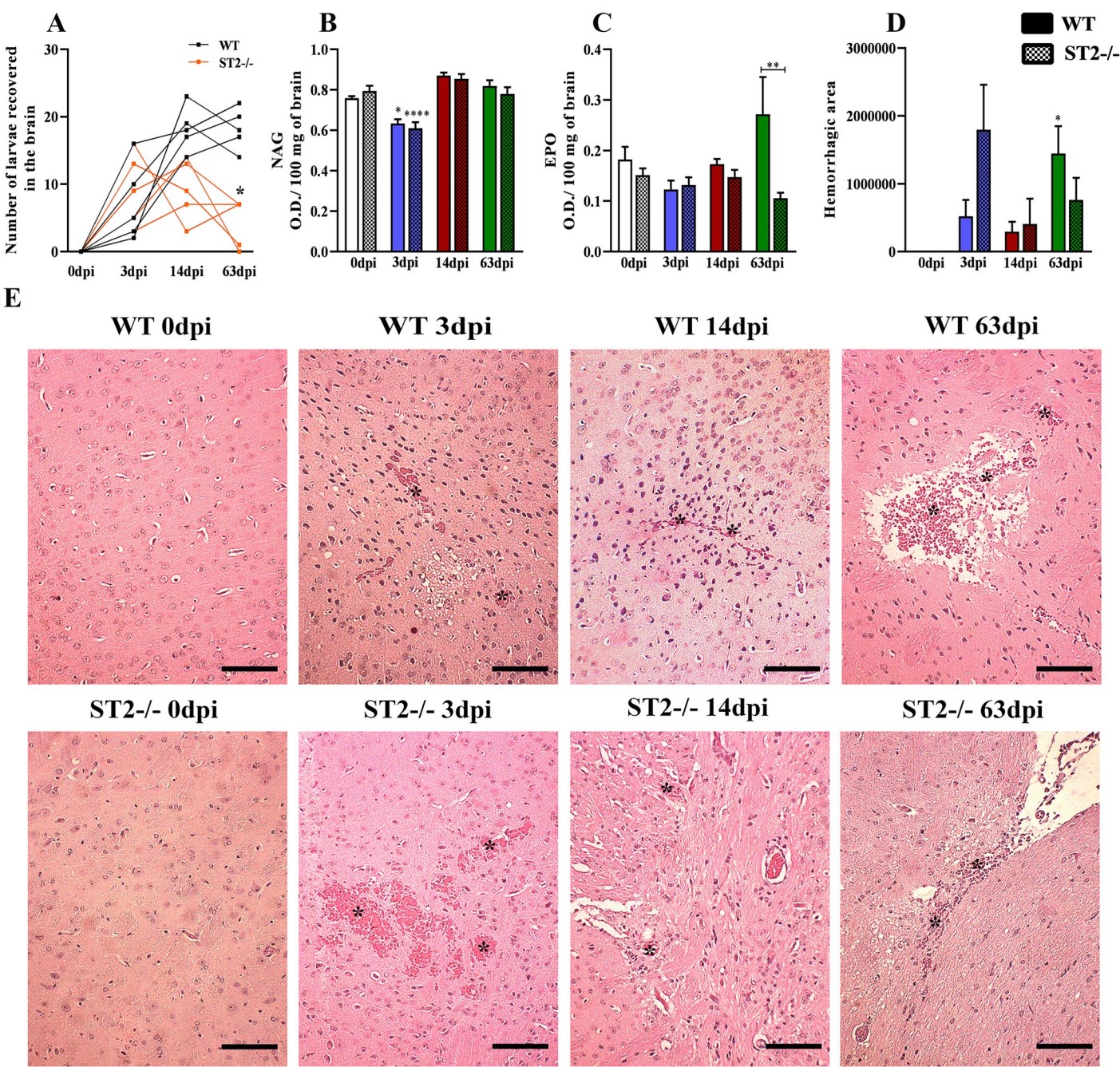

**Fig 8. Number of larvae recovered and tissue inflammation during infection in the brain.** (A) Number of larvae recovered in the brain; (B) N-acetylglucosaminidase (NAG) activity in brain tissue; (C) Eosinophils peroxidase (EPO) activity in brain tissue; (D) Hemorrhage score; (E) Representative hematoxylin and eosin staining of brain sections, hemorrhage area (*), Bar = 200µm. Results represent mean ± S.E.M., *$p < 0.05$, **$p < 0.01$, ****$p < 0.0001$. One-way ANOVA test and Kruskal-Wallis test followed by Dunn's test were used.

toxocariasis control [27,29]. In general, helminth infections develop a Th2-type immune response with an increase in IL-4, IL-5, IL-13 and IL-33 cytokines, which may assist in the expulsion of worms in the lungs [19], but there are still no studies on the importance of the cytokine IL-33 in *T. canis* infection. Thus, we thought of the hypothesis regarding the influence of the IL-33/ST2 pathway on the host immune response and its effects on *T. canis* migration.

Our results suggest that the IL-33/ST2 pathway in *T. canis* infection contributes to the increase of systemic eosinophils resulting in increased inflammation and tissue damage, in addition to reducing the Th17 response which may be related to the control of parasite migration to the brain.

During tissue migration, *T. canis* larvae release secretion and excretion products (TES) that promote leukocytosis with eosinophilia in their hosts [6–8]. However, the absence of the ST2 receptor reduced eosinophilia and the presence of monocytes in peripheral blood during infection. IL-33 act as an alarmin, recruiting chemokines that promote monocyte migration, in addition to having a role in signaling cell damage [30,31]. Studies have also indicated that IL-33 acts directly on eosinophils and regulates their biology, survival, activation and adherence [27].

The humoral response during infection by *T. canis* is the basis for the diagnosis and monitoring of disease treatment [2,8]. A study with *T. canis*-infected mice has reported an increase in IgG1, IgG2a, IgG2b and IgG3 in chronic phases of the disease, and IgG1 has been shown to be the best marker of infection [32], presenting findings similar to our study. In ST2$^{-/-}$ mice an increase in total IgG, IgG1, IgG2a and IgG2b was observed when compared to WT, suggesting that IL-33 may interfere with the humoral immune response. However, further studies relating the IL-33/ST2 pathway with the humoral response in *T. canis* infection are required.

After the larvae are released from the eggs, in the intestine, they penetrate the intestinal wall and migrate through the circulatory system to various organs, migrating first to the liver, then to the lungs and brain [4]. The *T. canis* larvae reach the liver with 1dpi, and in the initial stage of infection it is already possible to observe inflammatory foci dispersed throughout the liver parenchyma [6,33]. In our study, associated with a reduction in the parasite load in the liver tissue, the ST2$^{-/-}$ mice increased the hepatic inflammatory foci with 1dpi (S1 Fig), suggesting an early innate immune response in these animals or the influence of the gut microbiota. Studies show that there is a difference in the gut microbiota between different strains of mice, and the imbalance of this immune response can alter the individual's immunological fitness [34], thus, the gut microbiota of ST2$^{-/-}$ mice may be hindering the penetration of larvae into the gut mucosa, or the bacteria of the microbiota can translocate together with the larvae to the liver tissue, triggering a differentiated immune response in ST2$^{-/-}$ mice.

The presence of lesions in the liver parenchyma, granulomatous inflammatory response, increased ALT levels, presence of organized granulomas, and areas of liver necrosis during *T. canis* infection have also been demonstrated in other studies [6,35]. However, in ST2$^{-/-}$ mice there were smaller inflammatory areas with the presence of few eosinophils and a reduced number of granulomas. A combined role of macrophages and eosinophils has been proposed as key players in the mechanisms of granuloma formation during helminth infections [36]. IL-33 is related to the increase in eosinophils, and although this cell is considered important for the control of helminth infections, studies suggest that in *T. canis* infection, eosinophils may have a limited ability to kill these helminths and the larvae can escape efficiently from these cells. In addition, the inappropriate accumulation and activation of eosinophils can result in direct tissue damage through the release of highly cytotoxic granular proteins [27,37–40]. Thus, the reduction of eosinophils may be contributing to the ST2$^{-/-}$ mice presenting less hepatic inflammation and granuloma.

In several helminth infections, IL-4, IL-13 and IL-5 producing Th2 cells are related to the most important effector mechanisms in adaptive pulmonary immunity against helminths [41]. In our study, ST2$^{-/-}$ mice decreased the concentration of IL-5 and increased IL-33, IL-13 and IL-17 in the initial stage of infection, and with 63dpi in addition to IL-33 and IL-17, also increased IL-1β concentration. IL-33 acts directly on eosinophils and induces the production of type 2-associated cytokines from several types of cells, including Th2 cells, ILC2s, mast cells

and basophils, and consequently affects eosinophilic inflammation through the induction of IL-5, which is a cytokine known to activate eosinophils. Thus, the reduction of IL-5 in these animals may be directly related to the absence of the IL-33/ST2 pathway. Studies in animal models reveal that the role of IL-1β in type 2 immunity is complex and pathogen-specific, and that although this cytokine is known to promote Th1 and Th17 responses, it can also participate in Th2-mediated immunity [42,43]. Zaiss *et al.* [44] found that the nematode *Heligmosomoides polygyrus* is capable of inducing secretion of IL-1β in the intestine and suppressing the production of the innate cytokines IL-25 and IL-33, resulting in suboptimal type 2 immunity and allowing chronicity of the pathogen. Thus, it is suggested that there is a relationship between the IL-33/ST2 pathway and IL-1β, but its role in *T. canis* infection is not clear. Studies have been expanding the relation of IL-17 to helminth infections, Nogueira *et al.* [21] observed that mice after multiple exposures to *Ascaris suum* exhibited greater control of larval migration due to intense pulmonary inflammation associated with a polarized systemic Th2/Th17 immune response. Resende *et al.* [6] demonstrated that in toxocariasis there is a mixture of Th2 and Th17 inflammatory responses, observed by the increase of cytokines IL-4, IL-5, IL-13, IL-33 and IL-17 in the serum of mice during larval migration, showing that *T. canis* larvae are capable of triggering the Th17 response. There is evidence that there is a complex relationship between IL-17 and the cytokines of the Th2 response, while it can increase the production of IL-4 and IL-13 from Th2 cells and innate lymphoid cells, these same cytokines can also interrupt the production of IL-17 through a negative feedback loop [45]. Studies also indicate that there is a relationship between IL-33 and the Th17 response, one study reported that IL-33 was able to suppress IL-17A production by attenuating experimental autoimmune encephalomyelitis [46]. Other study also noted that administration of IL-33 suppressed the Th17 response secreted by lamina propria lymphocytes (LPL) and replaced the Th1 response with Th2 in chronic colitis induced by sodium dextran sulfate (DSS) in mice [47]. Therefore, further studies are needed to identify the relationship between the Th17 response and the IL-33/ST2 pathway during *T. canis* infection. Our observations show that during infection, the absence of the IL-33/ST2 pathway influences the pulmonary immunological response, mainly inducing the Th17 response.

Upon reaching the lungs, *T. canis* larvae triggered an inflammatory response in the lung parenchyma and in the airways, with the presence of hemoglobin and protein leakage, altering vascular permeability, as described by [48]. The ST2$^{-/-}$ mice showed a reduction in leukocytes in BAL, mainly in eosinophils, less perivascular inflammatory infiltrate, reduction in the number of cells and increase in macrophage infiltrate in pulmonary parenchyma and consequently generated a smaller number of granulomas. Studies showed that ST2$^{-/-}$ mice prompted by allergen-induced airway hyperresponsiveness (AHR) presented reduced inflammation in the peripheral airways, suggesting that IL-33/ST2 signaling is important for the influx of cells into the airways [49]. Townsend *et al.* [50] using a model of pulmonary granuloma formation induced by *Schistosoma mansoni* eggs, demonstrated that the formation of primary pulmonary granuloma is severely inhibited in ST2$^{-/-}$ mice, in addition to reducing eosinophils in the lungs, as occurred in our study. Thus, we believe that the IL-33/ST2 pathway influences the formation of granulomas during helminth infections and may be associated with eosinophils. However, it is important to note that a larger number of granulomas may not be associated with a reduction of parasitic load in *T. canis* infection, since as larvae are able to leave the confines of a fibrotic encapsulated granuloma, migrate to another place and begin the granulomatous process again [51].

Finally, when analyzing the pulmonary function in the initial phase of infection, substantial changes in pulmonary physiology were observed in ST2$^{-/-}$ mice compared to WT, probably due to the greater hemorrhagic condition and the type of cells present during inflammation,

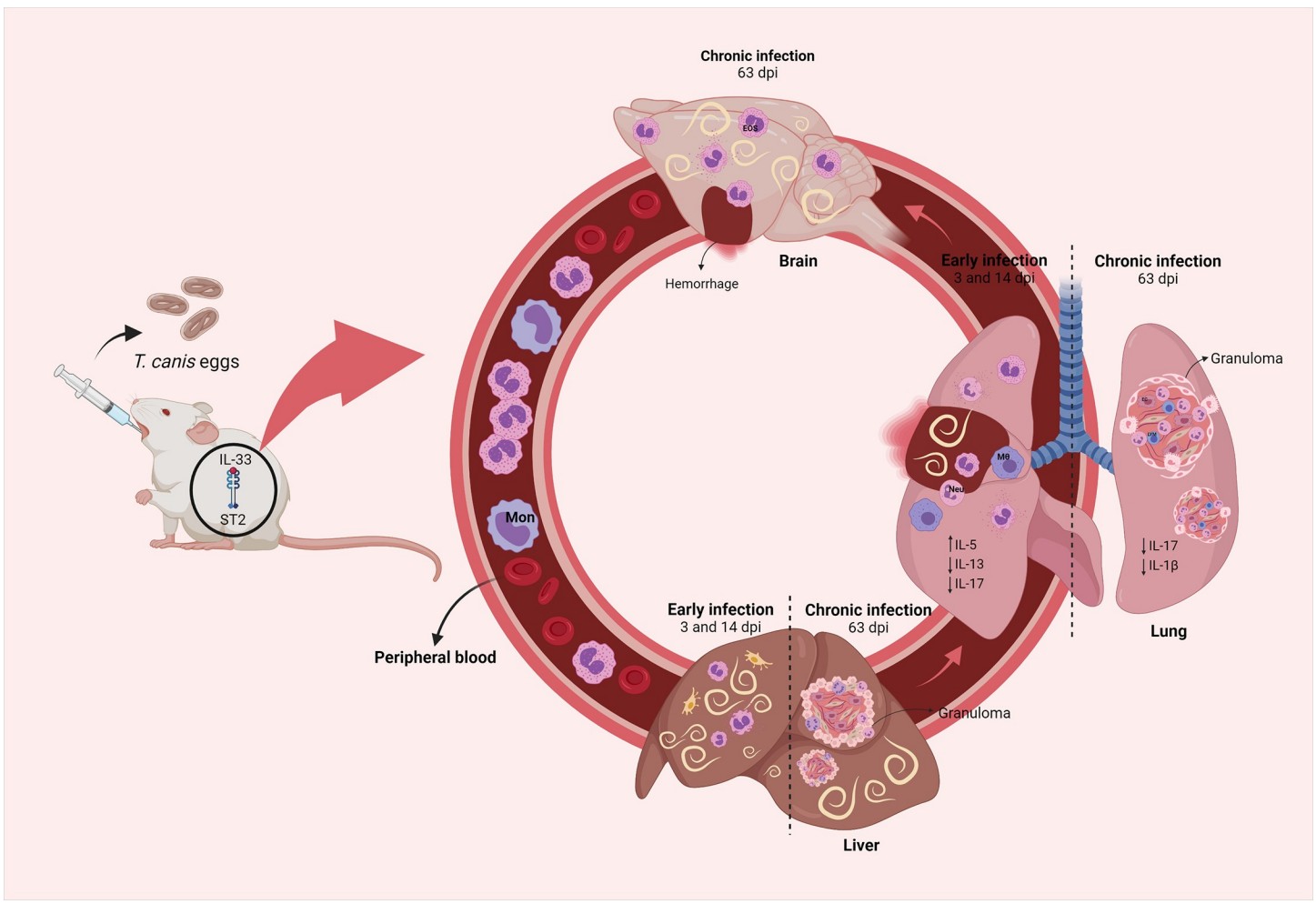

**Fig 9. Influence of the IL-33/ST2 pathway on *T. canis* infection.** The IL-33/ST2 pathway during *T. canis* infection influences the increase of hepatic parasite load and eosinophilia in the acute phase of the infection, causing greater liver damage and granulomas. In the lungs, it contributes to an increase in the Th2 response, reduces the Th17 response, increases eosinophils, tissue inflammation and the number of granulomas. And in the chronic phase in the brain, it contributes to an increase in the parasite burden and tissue eosinophilic activity.

showing greater macrophage activity by increasing NAG and reducing eosinophilic inflammation. The type 2 immune response in helminth infections takes several days, however, there is a rapid mechanism of tissue repair at the beginning of helminth infection, occurring by means of alternatively activated macrophages (AAMs), which can be rapidly induced by innate immune cells such as ILC2s, that are essential to limit damage and repair affected tissue [52]. Another factor to note is that eosinophils during some helminth infections, such as toxocariasis, release proteins that are sometimes harmful to the surrounding tissues of the host, intensifying tissue damage [52,53]. Thus, our study demonstrates that the IL-33/ST2 pathway is also capable of altering pulmonary physiology, worsening airflow in the airways.

After passing through the first migration phase that predominantly affects the liver and lungs, *T. canis* larvae migrate to muscles and the brain, this being known as myotropic-neurotropic phase [54]. Studies have shown that the larvae can remain in the brain tissue for long periods, being non-encapsulated and alive, suggesting advantages for the parasite to accumulate in an organ considered immune-privileged where they could be protected from the host lymphoid system [55,56]. In the present study, we observed that the $ST2^{-/-}$ mice showed a

decrease in the parasitic burden in the chronic phase of the infection, which may be associated with the type of local immune response, mainly due to the reduction of eosinophilic activity, which has been shown not to contribute to the control of burden parasitic. The systemic immune response and other tissues, such as the lungs, may also have contributed to preventing more larvae from migrating to the brain. However, we do not rule out the possibility that some larvae in these animals have a preference for migrating to other tissues, such as muscles, which were not studied in this work.

In conclusion, we describe the role of the IL-33/ST2 pathway during *T. canis* infection in the acute and chronic phase of the disease (Fig 9). The presence of the IL-33/ST2 pathway increases eosinophilia and eosinophilic activity in tissues, increases the hepatic and cerebral parasitic burden and contributes to the formation of hepatic and pulmonary granuloma, which does not control larval migration nor causes tissue damage. Also, this pathway reduces airflow in the airways and changes the immune response in the lungs, decreasing the macrophage and IL-17 activity that may be important for the control of tissue damage and infection. Our observations indicate that the presence of the IL-33/ST2 pathway induces susceptibility to toxocariasis, which could be a possible therapeutic target for study in the control of larval migration and prevention of tissue damage. Finally, further studies on the immunological and pathophysiological mechanisms are needed to understand the parasite-host relationship during *T. canis* infection.

## Supporting information

**S1 Fig. Liver inflammation score and representative hematoxylin and eosin staining of liver sections in WT and ST2$^{-/-}$ mice infected with *T. canis* with 1dpi.** (A) Liver inflammation score; (B) Representative hematoxylin and eosin staining of liver sections, hepatic parenchyma (P), vein lobular center (Vc), inflammatory infiltration foci (arrowheads). Bar = 200μm. Statistical comparisons were made between each strain with its specific uninfected group (0dpi) represented by the asterisk without the bar and between strains at the same time of infection represented by the asterisk with the bar. Results represent mean ± S.E.M., $^{**}p<0.01$, $^{****}p<0.0001$. One-way ANOVA test and Kruskal-Wallis test followed by Dunn's test were used.
(TIF)

**S2 Fig. Profile of cytokines present in lung tissue during infection with *T. canis*.** (A) TNF-α; (B) IFN-γ; (C) IL-1-ß; (D) IL-6; (E) IL-12/IL-23p40; (F) IL-17; (G) IL-4; (H) IL-5; (I) IL-33; (J) IL-13; (K) IL-10; (L) TGF-ß. Statistical comparisons were made between each strain with its specific uninfected group (0dpi) represented by the asterisk without the bar and between strains at the same time of infection represented by the asterisk with the bar. Results represent mean ± S.E.M., $^{*}p<0.05$, $^{**}p<0.01$, $^{***}p<0.001$, $^{****}p<0.0001$. One-way ANOVA test and Kruskal-Wallis test followed by Dunn's test were used.
(TIF)

**S1 Table. Main findings in the liver, lung and brain histopathological analysis in *T. canis* infection.**
(PDF)

## Acknowledgments

We are grateful to Silvana Tecles Brandão, manager of the Zoonosis Control Center of the city of Belo Horizonte and to the local veterinarians for their assistance in the collection of *T. canis* adult worms. Figures were made using biorender.com.

## Author Contributions

**Conceptualization:** Thaís Leal-Silva, Ricardo Toshio Fujiwara, Lilian Lacerda Bueno.

**Data curation:** Thaís Leal-Silva, Flaviane Vieira-Santos, Fabrício Marcus Silva Oliveira, Luiza de Lima Silva Padrão, Lucas Kraemer, Pablo Hemanoel da Paixão Matias, Camila de Almeida Lopes, Ana Cristina Loiola Ruas, Isabella Carvalho de Azevedo, Denise Silva Nogueira, Remo Castro Russo.

**Formal analysis:** Thaís Leal-Silva, Fabrício Marcus Silva Oliveira, Milene Alvarenga Rachid, Remo Castro Russo.

**Funding acquisition:** Ricardo Toshio Fujiwara, Lilian Lacerda Bueno.

**Investigation:** Thaís Leal-Silva, Flaviane Vieira-Santos, Fabrício Marcus Silva Oliveira, Lucas Kraemer, Lilian Lacerda Bueno.

**Methodology:** Thaís Leal-Silva, Flaviane Vieira-Santos, Fabrício Marcus Silva Oliveira, Denise Silva Nogueira, Milene Alvarenga Rachid, Marcelo Vidigal Caliari, Remo Castro Russo, Ricardo Toshio Fujiwara, Lilian Lacerda Bueno.

**Resources:** Milene Alvarenga Rachid, Marcelo Vidigal Caliari, Remo Castro Russo, Ricardo Toshio Fujiwara, Lilian Lacerda Bueno.

**Supervision:** Thaís Leal-Silva, Ricardo Toshio Fujiwara, Lilian Lacerda Bueno.

**Writing – original draft:** Thaís Leal-Silva, Flaviane Vieira-Santos, Fabrício Marcus Silva Oliveira, Lucas Kraemer, Denise Silva Nogueira, Milene Alvarenga Rachid, Marcelo Vidigal Caliari, Remo Castro Russo, Ricardo Toshio Fujiwara, Lilian Lacerda Bueno.

**Writing – review & editing:** Thaís Leal-Silva, Flaviane Vieira-Santos, Fabrício Marcus Silva Oliveira, Luiza de Lima Silva Padrão, Lucas Kraemer, Pablo Hemanoel da Paixão Matias, Camila de Almeida Lopes, Ana Cristina Loiola Ruas, Isabella Carvalho de Azevedo, Denise Silva Nogueira, Milene Alvarenga Rachid, Marcelo Vidigal Caliari, Remo Castro Russo, Ricardo Toshio Fujiwara, Lilian Lacerda Bueno.

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
