## [Decision Letter · Decision Letter 0]

25 May 2021

Dear Dr. Bueno,

Thank you very much for submitting your manuscript "Detrimental role of IL-33/ST2 pathway sustaining a chronic eosinophil-dependent Th2 inflammatory response, tissue damage and parasite burden during Toxocara canis infection in mice" for consideration at PLOS Neglected Tropical Diseases. As with all papers reviewed by the journal, your manuscript was reviewed by members of the editorial board and by several independent reviewers. The reviewers appreciated the attention to an important topic. Based on the reviews, we are likely to accept this manuscript for publication, providing that you modify the manuscript according to the review recommendations. 

Sincerely,

Subash Babu

Associate Editor

Sara Lustigman

Deputy Editor

Reviewer's Responses to Questions

**Key Review Criteria Required for Acceptance?**

**Methods**

-Are the objectives of the study clearly articulated with a clear testable hypothesis stated?

-Is the study design appropriate to address the stated objectives?

-Is the population clearly described and appropriate for the hypothesis being tested?

-Is the sample size sufficient to ensure adequate power to address the hypothesis being tested?

-Were correct statistical analysis used to support conclusions?

-Are there concerns about ethical or regulatory requirements being met?

Reviewer #1: (No Response)

Reviewer #2: -yes

-yes

-yes

-yes

-yes

-yes

Reviewer #3: Yes

**Results**

-Does the analysis presented match the analysis plan?

-Are the results clearly and completely presented?

-Are the figures (Tables, Images) of sufficient quality for clarity?

Reviewer #1: (No Response)

Reviewer #2: -yes

-yes

-yes

Reviewer #3: All the provide results are clearly presented

**Conclusions**

-Are the conclusions supported by the data presented?

-Are the limitations of analysis clearly described?

-Do the authors discuss how these data can be helpful to advance our understanding of the topic under study?

-Is public health relevance addressed?

Reviewer #1: (No Response)

Reviewer #2: -yes

-yes

-yes

-yes

Reviewer #3: Study findings are very well supporting the data.

**Editorial and Data Presentation Modifications?**

Reviewer #1: (No Response)

Reviewer #2: (No Response)

Reviewer #3: (No Response)

**Summary and General Comments**

Reviewer #1: This work from Leal-Silva et al. is a comprehensive comparison of the effect of a controlled T. canis infection in wildtype (WT)and ST2-knock-out (KO) mice. The ‘n’ values are of a good size (n=6-7), although I am a bit surprised by how tight some of the data are in the murine immune assays. This is a nice description of the impact of a global KO of the ability for IL-33 to signal through ST2.

The work makes a contribution to the field.

Attention to the following issues is needed.

A major concern is that the WT mice and KO mice will have different gut microbiotas and data are not omnipresent on how the gut bacteria effects host physiology/immunology (e.g. a single species difference – segmented filamentous bacteria – was critical to Th17 immunity). The advised way to do the experiments is to cross WT with KO, get the heterozygotes, cross them and then use WT and KO from the same litters. It is not reasonable to ask the authors to do this to repeat their study (although some comparison of KO mice (from San Paulo) with KO mice bred in this way would be advantageous). It is critical that the authors acknowledge this difference and state it clearly in the discussion as a possibility that could explain some of the differences between the two strains.

The discussion is too long (8 pages) and is mostly an expanded version of the descriptive results. This needs to be shortened and focused to highlight the point of this paper. If I get it right, that is, in this mouse model IL-33 signaling through ST2 exaggerates immunopathology to T. canis.

While noting the lengthy discussion, it would be of value to the reader for the authors to highlight that this is global ST2-KO, and then present some target cells in which absence of ST2 might play critical roles in the biology of T. canis infection.

With such a lot of descriptive data, perhaps a summary ‘carton’ figure would help the reader integrate and conceptualize the study.

Are all the data for a single experiment with 13 mice per time-point? If so, please state this clearly in the methods or results section.

Why are only female mice used?

Methods: were all the analysis blinded? This would be particularly important in the histological analysis.

Fig. 1: there appears to be a baseline (day 0) increase in leucocytes in ST-KO. This is worth commentary.

Fig. 2. If the * in panels a and b indicate a difference between WT and KO mice, I would be skeptical of these finding and doubt any biological relevance. The data points literally overlap.

Reviewer #2: The manuscript by Thaís Leal-Silva et al., is an important paper that not only confirms other studies showing e IL-33/ST2 pathway is involved in eosinophilia, hepatic and cerebral parasitic burden and induces the formation of granulomas related to tissue damage and pulmonary dysfunction, but advances the field by looking into mechanisms that might explain the effects of helminths. The work is of very high quality and has carefully analyzed and reported the data. However, I have few concerns 

1. Authors could include spaghetti plot giving better details for each subject at different time points

2. Authors could check for the legends and patterns in the graphs 

3. Authors could show the relationship between parasitic burden, antibody levels, leukocyte profiles and the cytokines levels to strengthen their hypothesis

4. Authors could include table for the comparison of histopathology in liver, lungs and brain from Wild-Type (WT) BALB/c

Reviewer #3: Comments

• Authors of the manuscript report that IL-33/ST2 pathway sustained the Th2 immune

response contributing to eosinophil activity, tissue damage and parasite tropism during infection by T. canis in mice. In this study authors have mechanistically also proven their hypothesis. Reported results are very interesting and novel.

• Authors have shown the Interleukin-33 (IL-33) related IL-1 family cytokines in the table format, for the better understanding it can be shown in graphical format. 

• It will be also good if authors could perform one or two cellular based experiments using the flow cytometry, it will further strengthen their current findings.

PLOS authors have the option to publish the peer review history of their article (what does this mean?). If published, this will include your full peer review and any attached files.

Reviewer #1: No

Reviewer #2: No

Reviewer #3: No

Figure Files:

Data Requirements:

Reproducibility:

References

---

## [Decision Letter · Decision Letter 1]

9 Jul 2021

Dear Dr. Bueno,

We are pleased to inform you that your manuscript 'Detrimental role of IL-33/ST2 pathway sustaining a chronic eosinophil-dependent Th2 inflammatory response, tissue damage and parasite burden during Toxocara canis infection in mice' has been provisionally accepted for publication in PLOS Neglected Tropical Diseases.

Best regards,

Subash Babu

Associate Editor

Sara Lustigman

Deputy Editor

Reviewer's Responses to Questions

**Key Review Criteria Required for Acceptance?**

**Methods**

-Are the objectives of the study clearly articulated with a clear testable hypothesis stated?

-Is the study design appropriate to address the stated objectives?

-Is the population clearly described and appropriate for the hypothesis being tested?

-Is the sample size sufficient to ensure adequate power to address the hypothesis being tested?

-Were correct statistical analysis used to support conclusions?

-Are there concerns about ethical or regulatory requirements being met?

Reviewer #2: -Yes

-Yes

-Yes

-Statistical analysis support the conclusion

-Yes

**Results**

-Does the analysis presented match the analysis plan?

-Are the results clearly and completely presented?

-Are the figures (Tables, Images) of sufficient quality for clarity?

Reviewer #2: -Yes

-Yes

-Yes

**Conclusions**

-Are the conclusions supported by the data presented?

-Are the limitations of analysis clearly described?

-Do the authors discuss how these data can be helpful to advance our understanding of the topic under study?

-Is public health relevance addressed?

Reviewer #2: -Yes

-Yes

-Yes

-Yes

**Editorial and Data Presentation Modifications?**

Reviewer #2: (No Response)

**Summary and General Comments**

Reviewer #2: (No Response)

PLOS authors have the option to publish the peer review history of their article (what does this mean?). If published, this will include your full peer review and any attached files.

Reviewer #2: No

---

## [Editor Report · Acceptance letter]

27 Jul 2021

Dear Dr. Bueno,

We are delighted to inform you that your manuscript, "Detrimental role of IL-33/ST2 pathway sustaining a chronic eosinophil-dependent Th2 inflammatory response, tissue damage and parasite burden during Toxocara canis infection in mice," has been formally accepted for publication in PLOS Neglected Tropical Diseases.

Best regards,

Shaden Kamhawi

co-Editor-in-Chief

Paul Brindley

co-Editor-in-Chief
